# Epidermal Microbiomes of Leopard Sharks (*Triakis semifasciata*) Are Consistent across Captive and Wild Environments

**DOI:** 10.3390/microorganisms10102081

**Published:** 2022-10-21

**Authors:** Asha Z. Goodman, Bhavya Papudeshi, Michael P. Doane, Maria Mora, Emma Kerr, Melissa Torres, Jennifer Nero Moffatt, Lais Lima, Andrew P. Nosal, Elizabeth Dinsdale

**Affiliations:** 1Department of Biology, San Diego State University, San Diego, CA 92182, USA; 2College of Science and Engineering, Flinders University, Bedford Park, SA 3929, Australia; 3Scripps Institution of Oceanography, Universtity of California, San Diego, CA 92093, USA; 4Department of Biology, Point Loma Nazarene University, San Diego, CA 92106, USA

**Keywords:** Chondrichthyes, microbiome, metagenome-assembled genomes, uncultivated microorganisms, captivity, leopard shark, biodiversity

## Abstract

Characterizations of shark-microbe systems in wild environments have outlined patterns of species-specific microbiomes; however, whether captivity affects these trends has yet to be determined. We used high-throughput shotgun sequencing to assess the epidermal microbiome belonging to leopard sharks (*Triakis semifasciata*) in captive (Birch Aquarium, La Jolla California born and held permanently in captivity), semi-captive (held in captivity for <1 year in duration and scheduled for release; Scripps Institute of Oceanography, San Diego, CA, USA) and wild environments (Moss Landing and La Jolla, CA, USA). Here, we report captive environments do not drive epidermal microbiome compositions of *T. semifasciata* to significantly diverge from wild counterparts as life-long captive sharks maintain a species-specific epidermal microbiome resembling those associated with semi-captive and wild populations. Major taxonomic composition shifts observed were inverse changes of top taxonomic contributors across captive duration, specifically an increase of Pseudoalteromonadaceae and consequent decrease of Pseudomonadaceae relative abundance as *T. semifasciata* increased duration in captive conditions. Moreover, we show captivity did not lead to significant losses in microbial α-diversity of shark epidermal communities. Finally, we present a novel association between *T. semifasciata* and the *Muricauda* genus as Metagenomes associated genomes revealed a consistent relationship across captive, semi-captive, and wild populations. Since changes in microbial communities is often associated with poor health outcomes, our report illustrates that epidermally associated microbes belonging to *T. semifasciata* are not suffering detrimental impacts from long or short-term captivity. Therefore, conservation programs which house sharks in aquariums are providing a healthy environment for the organisms on display. Our findings also expand on current understanding of shark epidermal microbiomes, explore the effects of ecologically different scenarios on benthic shark microbe associations, and highlight novel associations that are consistent across captive gradients.

## 1. Introduction

The eukaryotic organism harbors an astounding number of microbes both internally and externally to supplement nutrient breakdown, thwart pathogenic colonization, and provide defense via molecular byproducts, and is known as the microbiome. The eukaryotic host and the respective microbiome has been reclassified as a metaorganism, representing a functional system that adapts to changing environments and external threats by cultivating a synergistic interdependence with microscopic symbionts [1]. The epidermal microbiome of the marine metaorganism reflects both environmental conditions and host-microbiome interdependent effects, in addition to directly interfacing with seawater, while remaining distinct from the pool of microbes in the water column [2,3,4,5]. The epidermal microbiomes belonging to triple fin fish (*Forsterygion capito*) experiencing polluted conditions, for example, have lower levels of biodiversity compared to fish residing in pristine environments [6]. Likewise, microbial profiling of stranded leopard sharks (*Triakis semifasciata*) revealed infections caused by the pathogenic bacteria *Miamiensis avidus*, causing the mass die-offs of *T. semifasciata* populations in the San Francisco Bay the summer of 2017 [7]. While the presence of unicellular species is telling, metabolic potentials of epidermal microbiomes is equally descriptive of environmental conditions. The epidermal microbiomes belonging to thresher sharks (*Alopias vulpinus*) encoded higher levels of genes associated with heavy metal resistance [2,3] indicating an increased supply of cobalt, zinc, cadmium, and iron from the host, as cartilaginous (Chondrichthyan) fishes are known to retain heavy metals in the integumentary tissues [8]. Therefore, the taxonomy and functional profiles of epidermal microbiomes serve as indicators of environmental conditions, disease, and the provision of host-supplied nutrients. To this point, data for microbiomes belonging to captive, marine vertebrate hosts is still lacking [9], and is scarcer for the Chondrichthyan clade.

Extant sharks (Chondrichthyes subclass: Elasmobranchii, superorder: Selachimporhpa) are vertebrates showcasing an ancient host-associated microbiome in addition to harboring diverse biophysical and physiological traits such as the ability to naturally produce bioluminescence via photophores [10] and the ability to detect the electromagnetic fields produced by all living organisms using ampullae of Lorenzini [11]. The shark epidermis, in particular, is exceptional due to the presence of overlapping dermal placoid scales (denticles) connected to numerous, horizontally arranged layers of muscles that are known to contract, thereby optimizing the build-up of hydrostatic pressure during fast swimming [12,13]. The physiology of the shark epidermis supports protective functions similar to those of Ray-fin fish (Actinopterygii), whereby continuous shedding of denticles, in concert with low drag and the flexion of the scales, reinforces antifouling efforts. The shark epidermis is not completely free of biofilms; although elasmobranchs produce a reduced mucosal layer relative to teleost fish, several species including *T. semifasciata*, *A. vulpinus*, the whale shark (*Rhincodon typus*), and the round ray (*Urobatis helleri*) have been observed to have a microbiome that mirrors the respective evolutionary trajectory of the host [3,14]. This concept by which the microbiomes reflect host phylogeny is called phylosymbiosis. The combination of the physical and chemical barrier that is the mucosal layer, acting jointly with a maintained microbiome across evolutionary time, may account for the few reported cases of infections witnessed in wounded, wild black-tip reef shark (*Carcharhinus melanopterus*) populations [15]; for each shark, pathogens were absent in the unchanged epidermal microbiome sampled from the wound site. Investigations into shark-associated epidermal microbiomes, therefore, are necessary to supplement the scarcity of information concerning the shark metaorganism.

While host-related factors including biological sex, health status [16], and diet [17] regulate microbial recruitment and retainment, environmental factors such as geography, pH, and salinity also contribute to the composition of microbial communities and functions [18,19]. Public aquariums are multipurpose as they (1) present a range of organisms to the public to promote education and conservation, (2) are often involved in breeding programs for endangered species, and (3) aim to promote healthy living conditions for the captive organisms, e.g., enhance longevity and reduce disease outbreaks. To achieve a reduced host pathogen load, aquariums maintain consistent conditions with limited introduction of natural microbial communities due to water filtration, stable macroscopic species interactions, regular diets, and lower variations of abiotic factors. An outstanding question is whether the aquatic hosts kept under these conditions have maintained epidermal microbiomes similar to wild counterparts and provide host benefits. Of the 1370 species of Elasmobranches, only a small percentage of associated microbial communities have been characterized including those previously mentioned in addition to the nurse (*Ginglymostoma cirratum*), lemon (*Negaprion brevirostris*), sandbar (*Carcharhinus pleumbeus*), caribbean reef (*Carcharhinus perezii*), and tiger (*Galeocerdo cuvier*) sharks and the thornback (*Raja clavate*) and round (*Urobatis halleri*) stingrays [2,3,14]. A single study of a captive population was performed with cownose rays (*Rhinoptera bonasus*) exclusively residing in an aquarium touch tank [20]. Herein, we investigate the impact that a captive environment exerts on the epidermal microbiota associated with captive *T. semifasciata* individuals and compare these microbial communities to the composition and functional potentials of microbiomes belonging to wild sharks that live in nearshore habitats and temporarily held sharks representing semi-captivity.

The *T. semifasciata* species was chosen as our model to investigate the change in microbiome composition across environments because they are endemic to the west coast of the USA, are an important component of the kelp forest ecosystem, and aggregate in large numbers in bays in in Southern California [21], where they are visible to local communities and accessible for sampling. In addition, these sharks are small, feed on benthic organisms, and are successfully kept in display settings, unlike larger sharks. These characteristics make them a popular item for aquarium. The four captive sharks were housed in a public aquarium that houses *T. semifasciata* for lifelong durations, and four semi-captive specimens were sampled in a research facility (held in captivity for <1 year in duration and scheduled for release). Further, the water for the two aquarium facilities is procured from the location where some of the wild leopard sharks were sampled. Several of the wild *T. semifasciata* sampled in the study were collected near shore to the aquarium housing the sampled captive sharks. All sharks were sampled for shotgun metagenomic analysis of their skin microbiomes. Our analysis explored the microbiome of sharks kept in a real-world situation, where only four leopard sharks were housed by the aquarium, providing appropriate conditions for the sharks and diversity for the viewing public. Sampling the microbiome from the skin is non-invasive and could provide an accessible and consistent sample location to monitor the health of captive sharks in the future. We hypothesize the taxonomic composition of the epidermal microbiomes belonging to *T. semifasciata* will remain consistent between captive and semi-captive sharks and wild peers. Furthermore, while the broad functional genes are expected to remain largely unchanged across captivity status, a few key functions are anticipated to differ as a reflection of the provisioning that occurs in captivity. Last, we will identify the presence of novel microbes that are consistent across wild, semi-captive, and captive sharks by constructing metagenome assemble genomes.

## 2. Materials and Methods

### 2.1. Sampling of Metagenomes

The epidermal microbiomes of wild *T. semifasciata* were sampled in the summers of 2013 (*n* = 6) and 2017 (*n* = 8) at La Jolla Shores in San Diego, CA, USA (32.868342, −117.255304), and the summer of 2015 (*n* = 5) at Moss Landing in Monterey County, California (36.801588, −121.791970). Wild sharks were sampled during the summer aggregations characteristic of *T. semifasciata* populations [21,22]. We targeted a minimum of three individual sharks per captivity group. Wild sharks were initially caught using a handline with baited barbless circle hook and, using a scoop net, brought onboard for sampling and subsequent release. Four *T. semifasciata* (*n* = 4) were sampled in short term captivity (< 1 year) during the summer of 2016 while cohoused in a tank at the Scripps Research Institute of Oceanography in San Diego, California (32.865402, −117.252945). Captive *T. semifasciata* (*n* = 4) were sampled in the summer of 2018 at the Birch Aquarium at Scripps Institution of Oceanography in La Jolla, California (32.865658, −117.250535) to represent microbiomes belonging to *T. semifasciata* long held (>1 year) in captivity. Captive sharks were corralled by divers into a pen and immobilized in a sling during sampling.

To consistently describe the shark epidermis-associated microbiome, microbes were collected between the pectoral fin and dorsal fin along the lateral line on the left side of the organism (Figure 1A) using a blunt, closed-circuit syringe. The syringe is prefilled with filtered seawater and flushed against the epidermis to dislodge and collect microbes reducing introduction of environmental microbes and host cells (Figure 1B, [2]). The approximate 200 mL of captured microbe was collected on a Sterivex (Millipore, Inc.), where one Sterivex per individual was obtained each collection date. The blunt edge syringe has been used many times to collect microbes from underwater organisms [2,3,4,23] and has the advantage of dislodging the microbiome from in and under the dermal denticles, without the disrupting the host tissue and becoming inoculated with products such as melanin [2,3,5,24].

To describe the water-associated microbial communities, bulk water samples were taken at each location: ~60 L of tank (*n* = 3) or ocean (*n* = 3) water were simultaneously collected and filtered once through a nylon mesh sieve (200 µm pore size) to remove large debris and eukaryotic organisms and then using tangential flow filtration (TFF, 100 kDa) to further concentrate the microbial consortia [25]. The resulting sample of approximately 500 mL composed the concentrated sample and was processed and collected on a Sterivex filter (0.22 µm).

### 2.2. DNA Extraction and Metagenome Sequencing and Annotation

Upon collection, cells were lysed within the Sterivex filters and purified using a Macherey Nagel Tissue Kit, (MACHEREY-NAGEL, Düren, Germany)as we have done previously [26]. Microbiome DNA was then prepared for shotgun metagenomic sequencing using the Swift 2S Plus Kit (Swift Biosciences, Ann Arbor, MI, USA). Purified DNA was sequenced using an Illumina MiSeq sequencer (Illumina, Inc., San Diego, CA, USA) with MiSeq v3 Reagent Kit (Illumina, Inc., San Diego, CA, USA) on site at San Diego State University as previously performed [2,19,27]. Following sequencing, reads were processed for quality control via PrinSeq to remove artificial duplicates, and reads having >10 N’s and <60 bp as per previous metagenomic workflows [28]. High quality, paired reads were uploaded to Metagenomic Rapid Annotations using Subsystems Technology (MG-RAST) to call taxonomic and functional gene assignments. MG-RAST accomplishes gene calling and protein prediction with BLAST comparisons to the NCBI and SEED protein databases [29], similar to previous analysis [25]. Annotated reads belonging to eukaryotic organisms were removed prior to data analysis. Sequencing annotations were conducted using the following parameters: e-value > 10^−5^, 70 percent identity, and >60 bp alignment length. The sequences with the highest bit score were reported as we have done previously [2,4,23,30]. The taxonomic annotation outputs were filtered in MG-RAST to only include those from Bacteria and Archaea domains. Viruses and Eukaryotes were not used in the analysis because they were underrepresented in the metagenome. Metagenomes were compared using proportional abundance, which is preferred to rarefaction [31,32,33].

### 2.3. Assembly and Annotation of Metagenome-Assembled Genomes

Metagenome-assembled genomes (MAGs) were constructed to identify whether metagenomes from the sharks across captivity and wild environments contributed sequences to the genomes, thus identifying shark specific microbes. The MAGs were generated from metagenomes belonging to 27 individual *T. semifasciata* across captivity status using MEGAHIT (v1.2.9) [34]. Contigs >1000 bp were run through METABAT2 (v2.15) binning program to reconstruct MAGs [35]. CheckM was applied to each MAG and for the remainder of this article, only MAGs with at least 80% completeness and less than 10% contamination (confirmed via CheckM; [36,37,38]) will be presented and discussed following quality control metrics outlined in previous investigations [39,40]. Bacterial delineations of MAGs were compared across several taxonomic classification algorithms including FOCUS [41], CheckM [36], and the Pathosystems Resource Integration Center (PATRIC; [42] and confirmed by comparing average nucleotide identity (ANI; [43]) or DNA Kmer-based evidence of similarity surpassing 90%, reflective of standard criteria [44]. We used the minimum information for metagenomic assembled genomes to classify the level of completeness and contamination to identify the quality of the MAGS. Of the 54 bins, eight met baseline requirements for further analysis, with three MAGs met the requirements to be classified as high quality MAGS (>90% completeness, <5% contamination) and three identified with medium quality MAGS>50% completeness, [45]). We utilized the advanced analysis and visualization tool *Anvi’o* to visually explore and interpret our assembly based metagenomes [46]. Functional gene annotation of MAGs was performed through PATRIC using the BLAT alignment tool (v35.1).

### 2.4. Statistical Analyses

To explore the microbiomes of shark across captivity status, first we compared the shark and water column microbiomes using permutation multivariate analysis of variance (PERMANOVA) analysis and then narrowed the investigation to the variation between shark microbiomes across captivity status using diversity measures and differences in distribution of taxon and functional potential. The α-diversity indices of microbial community richness, evenness, and diversity were measured using Margalef’s (*d*), Pielou’s (*J’*), and Inverse Simpson’s (1/λ) indices, respectively [47,48,49] to discern the effect captivity exerts over associated microbial community biodiversity. To measure the intraspecific similarities, and between group dissimilarities, similarity percentages breakdowns (SIMPER) were calculated [50]. To identify whether the taxonomy of the microbiome remained consistent or varied over captivity status, a permutation multivariate analysis of variance (PERMANOVA) on the relative abundance of each taxon level (from Order to genera) were conducted [31]. To produce relative abundance of each taxon level, the data sets were normalized to the sum of all taxa counts for each epidermal microbiome sample. PERMANOVA are designed for non-parametric data, particularly those where there is a larger number of variables compared with samples [51]. All data was fourth root transformed [4] which balances the effects of a community structured on a few abundant species and a community structured on all species, and thereby influenced by the occurrence of the rarest taxa [52]. The PERMANOVA used 999 random permutations. Microbial family levels were explored in detail because it has been identified that microbial communities converge to similar family level structures even as the species level vary [53]. Thus, several species belonging to the same metabolic family coexist and support stable growth of rare (<1% relative abundance) and more abundant taxa [53]. To measure the similarities between captivity groups, similarity percentages breakdowns (SIMPER) were calculated [50]. Association mapping between metagenomes across environments were visualized via non-metric multidimensional scaling (nMDS) generation with correlation overlays [54]. Tests for differences in group dispersion or homogeneity of the multivariate variations were generated using a PERMDISP analysis [55,56]. To account for the imbalance of samples between the wild (*N* = 19) and captive and semi-captive (*N* = 4), we randomly assigned wild metagenomes into ten groups of four and ran a pairwise PERMANOVA analysis and is included in Table A2.

We used metagenomics to describe the abundance of genes found by microbiome as a proxy for gene expression: although it does not measure which functional genes are being expressed at the point the sample was taken, it measures which functional genes are important for the bacteria in that environment [57]. There is a high level of correlation between the metagenomes and meta-transcriptomes [58], where the abundance of a gene in metagenomes is a predictor of its expression level in the meta-transcriptome and areas where the two analyses vary are associated with short term changes in expression rather than bacteria functions that are under strong selective pressure and are well adapted to their environment [59,60]. All functional data was investigated by comparing the proportion of sequences belonging to each metabolic group. The SEED’s Subsystem Annotation arranges functional pathways into a hierarchal structure, ranging from the broadest metabolic pathways (Level 1) to increasingly specific gene functions, with growing complexity denoted in between (Level 2 & 3) as resolution increases. This allows for mapping of key, obscure biochemical faculties upstream to encompassing parent pathways. For example, within the broadest metabolic group (Level 1) “respiration” is the gene functions associated with ATP synthase proteins. Therefore, the functional potential (across all SEED subsystem levels) of the metagenomes of wild, semi-captive and captive leopard sharks was tested using a PERMANOVA. Differences in metabolisms, as tested using an analysis of variance (ANOVA) with a post hoc Tukey test, were conducted and visualized on the Statistical Analysis of Metagenomic Profiles (STAMP) package (v2.1.3; https://beikolab.cs.dal.ca/software/STAMP) [61]. Statistics were run using the Primer-e package 7 (v7.0.2; www. primer-e.com/permanova.html, accessed on 28 January 2021) with the PERMANOVA+ add on, STAMP software [61], and GraphPad (v9.4.1; https://www.graphpad.com) PRISM 9 (v9.1.2). All graphs were generated using PRISM 9 and Anvi’o (v7; https://merenlab.org/software/anvio).

## 3. Results

From the epidermal microbiomes of 27 *T. semifasciata* individuals, 31,114,584 sequenced reads were identified as bacteria and archaea spanning 27 phyla, 42 classes, 92 orders, 208 families, and 564 genera (Table 1). The total number of families present for *T. semifasciata* metagenomes ranged from 164 to 208 families. Rarefaction curves show a similar trend in biodiversity (Appendix A
Appendix A). Water-associated microbiomes significantly differed from host-associated microbiomes in taxonomic composition (PERMANOVA: Family, Pseudo-F_df=1, 31_ = 2.056, P (perm) < 0.05). The water column microbial communities had little within-group variation (SIMPER analysis; 72.96 similarity) and high dissimilarity. The water column microbial communities, when paired against the epidermal microbiomes of each environment significantly differed (*p* < 0.05) between each other and against the epidermal microbiomes (*p* < 0.05) for each environment and are not compared further.

Microbial community richness (*d*), evenness (*J’*), and overall diversity (1/λ) were similar across captivity status (Table 2). To understand the taxonomic composition and variation across environments we then characterized each group of metagenomes. The major contributors to taxonomic classes of epidermal microbiomes belonging to *T. semifasciata* long held in captivity were Alteromonadales (57.1% ± 4.29 S.E.), Burkholderiales (5.91% ± 0.79) and Sphingomonodales (5.51% ± 0.85), belonging to Gamma-, Beta-, and Alphaproteobacteria clades respectively. The epidermal microbiomes of *T. semifasciata* inhabiting the southern coast of California harbored different abundances of major clades than their captive counterparts; finer-scale taxonomic resolution revealed differences between captive and wild benthic shark microbiomes were accounted for by major taxonomic contributors (>1% relative abundance, Figure 2), with smaller populations having no significant impact on microbiome compositions. While the Pseudoalteromonadaceae family dominates the skin microbiomes of *T. semifasciata* residing in captivity (21 ± 1.5 S.E.M.), the Flavobacteriaceae family contributed the most to metagenome compositions belonging to wild *T. semifasciata* (10% ± 2.0), followed by Pseudomonadaceae (8.9% ± 1.87) and Alteronomonadaceae (7.3% ± 0.94). The Pseudomonadaceae (19.0% ± 2.87) and Flavobacteriaceae (8.4% ± 0.34) families were among the top three major taxonomic contributors to *T. semifasciata* microbiomes for semi-captive shark microbiomes, with the Moraxellaceae (11.0% ± 2.1) family represented significantly (*p* < 0.001) more in the microbial composition.

No statistical differences were observed between wild and neither captive nor semi-captive epidermal microbiomes (Pairwise PERMANOVA Family level, t(3,18) < 1.3, P (perm) > 0.1); Genus level, t(3,18) < 1.3, P (perm) > 0.1). However, as is shown in Figure 2, there are several microbial families that vary in relative abundance across captivity status and these were identified in the SIMPER analyses where microbiomes belonging to captive sharks are more similar to those associated with semi-captive sharks (Bray–Curtis average similarity = 88) than to wild sharks (Bray–Curtis average similarity = 84), and semi-captive sharks are more similar to captive sharks than to wild sharks (Bray–Curtis average similarity = 86). The microbial families that were driving the differences between captive and wild *T. semifasciata* was attributed to higher abundances of Pseudoalteromonoadaceae in captive microbiomes, compared with the other two groups (3.0 Diss/S.D., Table 3). Taxonomic families accounting for consistent differences between semi-captive and wild sharks were Alcanivoraceae (2.8 Diss/S.D.), Planctomycetaceae (2.5 Diss/S.D.), and Moraxallanceae (2.4 Diss/S.D.), and were higher in proportional abundance in semi-captive metagenomes. The constant contributor of dissimilarity between captive and semi-captive individuals was attributed to Halomonodaceae (3.7 Diss/S.D.), which was approximately twice the relative abundance in semi-captive epidermal microbiomes. There was no difference in the microbiome variation of *T. semifasciata* epidermal microbiomes residing in each environment (PERMDISP: Family, Pseudo-F_df =2, 27_ = 1.29, P (perm) > 0.1, Genus, Pseudo-F_df =2, 27_ = 1.85, P (perm) > 0.1, Table 4, Table A1). The microbial families of *T. semifasciata* epidermal microbiomes did not form distinct clusters for each environment when visualized using an nMDS; the structure of *T. semifasciata* epidermal microbiome data across captivity showed no difference in coefficients of similarity and several microbial families were found to be highly correlated (>0.85; Figure 3 and Appendix A). These families include Pseudomonadaceae, Shewanellaceae, and Sphingomonodaceae, and represent major taxonomic contributors (>1%) for all three groups of *T. semifasciata*. Less represented clades (<1%) related to the spread of metagenomes include Beijerinckiaceae, Brucellaceae, Rickettsiaceae, and Bacteroidaceae as illustrated in nMDS overlays (Figure 3).

### 3.1. Comparisons of Functional Gene Potentials of T. semifasciata Epidermal Microbiomes across Environments

There were between 661 and 778 metabolic categories identified in all *T. semifasciata* metagenomes. Water column metagenomes functional potentials were significantly distinct from *T. semifasciata* epidermal microbiomes (PERMANOVA: subsystem level 1, Pseudo-F_df = 1, 31_ = 1.93, P (perm) < 0.05) and were not compared further.

Of the 27 major gene pathways identified in *T. semifasciata* metagenomes, six contributed >50% of reads, with the two top contributors (>10%) involving carbohydrate (12.0% ± 0.2 S.E.M.) and amino acid (11.0% ± 0.15) metabolism. Analyses of functional levels revealed no statistical differences between all groups (PERMANOVA: subsystems level 3, Pseudo-F_df 2, 27_ = 0.859, P (perm) > 0.5, Table 4). In addition, differences in similarity between groups were insignificant, as wild metagenome metabolic profiles were comparable (SIMPER: wild vs. semi-captive, Bray–Curtis Similarity = 90.02.; captive vs. semi-captive, Bray–Curtis Similarity = 91) with the greatest dissimilarity measured between wild and semi-captive shark populations (Bray–Curtis Similarity = 90) when analyzing metabolic pathways (SEED subsystem level 2, Figure 4). The functions contributing consistently to differences between wild and semi-captive epidermal microbiomes, as identified by SIMPER analysis, were genes involved in gene transfer agents (2.0% contribution, 2.0 Diss/S.D.), motility and non-flagellar swimming (1.7%, 1.4 Diss/S.D.), and protein secretion system type VII (1.5%, 1.4 Diss/S.D.).

Although β-diversity analyses revealed no statistical differences between functional profiles of metagenomes belonging to *T. semifasciata* epidermal microbiomes across captivity states, several specific functional pathways (SEED subsystems Level 2 and 3) were found to differ. Captive shark metagenomes featured significant (*p* < 0.001) increases in relative abundances of genes; in carbohydrate synthesis (14.0% ± 0.24 S.E.M.) and utilization, such as the more specific fermentation (0.96% ± 0.06, Figure 4) pathway (SEED subsystem: level 2). Specific functional pathways (SEED subsystem: level 3) featuring utilization of simple sugars and sugar alcohols were also significantly represented in captive metagenomes relative to other groups, including fructose and mannose inducible phosphotransferase system (PTS; 0.46% ± 0.01, *p* < 0.001), methylglyoxal metabolism (0.37% ± 0.01, *p* < 0.001), and mannitol metabolism (0.19% ± 0.26, *p* < 0.001, Figure 5). Genes coding for enzymes involved in the breakdown of these saccharides, i.e., beta-glucosidase (0.48% ± 0.01, *p* < 0.001), were correspondingly increased in captive shark epidermal metagenomes compared to semi-captive and wild counterparts, in addition to the subsequent alcohol synthesis including acetone, ethanol, and butanol (0.20% ± 0.09, *p* < 0.001). Pathways involved in heavy metal acquisition were likewise significantly increased in captive shark epidermal microbiomes, as indicated by increased genes involved in heme/hemin uptake and utilization (0.19% ± 0.03, *p* < 0.001), iron acquisition (2.7% ± 0.17, *p* < 0.001). Furthermore, pathways involved in virulence, disease, and defense were significantly more abundant in captive shark epidermal microbiomes (5.9% ± 0.03, *p* < 0.001), including specific genes related to periplasmic stress (0.02% ± 0.01, *p* < 0.001), capsular polysaccharide biosynthesis and assembly (0.13% ± 0.013, *p* < 0.001), murein hydrolytic activities (0.34% ± 0.01, *p* < 0.001), and the antibiotic resistance gene BlaR1 family regulatory sensory-transducer disambiguation (0.34% ± 0.32, *p* < 0.001) than semi-captive and wild populations. Conversely, significantly higher functional potentials involved in vitamin synthesis (7.2% ± 0.145, *p* < 0.001) and nitrogen metabolism (1.8% ± 0.001, *p* < 0.001) were observed in semi-captive samples. For *T. semifasciata* individuals, no significant gene pathways differed between semi-captive or wild metagenomes.

### 3.2. Metagenome-Assembled Genomes Constructed from Microbial Communities Associated with T. semifasciata

Cross assembly of the 27 *T. semifasciata* metagenomes yielded 54 MAGs containing 241 814 contigs greater than 1 kilobase pairs, with N50 of 735 bp and N75 of 583 bp. Of these, nine high quality MAGs were constructed spanning seven known bacterial phyla. The contribution of annotated MAGs from each group ranged from 5.5% ± 1.8 S.D. from captive hosts to 87% ± 0.66 from wild individuals, with semi-captive contributing 7.5% ± 1.1 to MAG generation, comparable to animal MAGs [62]. While all *T. semifasciata* metagenomes were involved in MAG assembly, three groupings of contributions to MAG generation can be distinguished in Figure 6, where more even mean coverage of MAG contribution by *T. semifasciata* populations across environment is visualized. Among the groups, group 1 in Figure 6 highlights heavy contributions from both captive and wild metagenomes, while not harboring any of the nine high quality MAGs, while group 2 contains two MAGs further investigated. Finally, group 3 has an even spread of mean contribution from each metagenome environments and contains three MAGs. The number of contigs wild shark hosts contributed to MAG assembly was greater (87.0% ± 4.9 S.E.M.) than both semi-captive (7.5% ± 6.9) and long-held captive sharks (5.5% ± 4.5) and is due to number of reads in the metagenomes.

Of the nine qualifying MAGs, two (Bin 27 and Bin 9) were identified as belonging to the *Muricauda* genus, an increasingly classified child taxon of Flavobacteriaceae family. Bin 27 had an average nucleotide identity match of 99.8% with the *Muricauda antarctica* species [43], while identification of the species of Bin 9 remains incompletely verified (<90% similarity) at species level, with the highest resemblance matched to *Muricauda reustringensis* (84% similarity). The DNA G+C content of Bin 27 and Bin 9 was 45.2% and 41.7%, respectively, both falling within the acceptable range reported for taxa belonging to the *Muricauda* genus, i.e., 41–45.4 mol% [63]. Bin 27 featured a 95.7% complete genome, with 4 106 genes, 4.2% genome redundancy, and a total length of 4 285 655 bp. The Bin 9 MAG was calculated to have a 91.6% complete genome composed of 4 673 genes, with more genomic redundancy (7.0%) and a longer total genome length of 4 625 638 bp (Figure 7). Following genome annotation, no resistance or susceptibility to antibiotics were found in Bin 27 encoding for *Muricauda antarctica*. However, two virulence features were discovered: GTP-binding and nucleic acid-binding protein YchF (fig|1055723.17.peg.1964), and ferric uptake regulation protein FUR (fig|1055723.17.peg.1945).

Of the remaining seven MAGs, two were of unknown taxonomic origin with no similar genomes (Bin 36 and 47). In decreasing order of confidence, the five remaining MAGs were identified as *Zunongwangia atlantica* (99.8%, Bin 30), *Roseivirga pacifica* (97%, Bin 30), *Leeuwenhoekiella blandensis* (64%, Bin 22), *Micavibrio spp.* (39%, Bin 13), and *Pseudomonas spp.* (34%, Bin 12; Table A1). Finally, Bin 36 most closely resembled a member of the *Fluviicola genus* (23% similarity), while Bin 47 matched most (20% similarity) to the *Thalassospira* genera.

## 4. Discussion

Captive environments often aim to conserve populations while balancing a complex ecosystem of micro- and macro-organisms. To help with these efforts we provide the first comparisons of taxonomic compositions and functional potentials between metagenomes associated with captive, semi-captive and wild *T. semifasciata* populations and show that captive environment exerts no detrimental pressure to *T. semifasciata* epidermal microbiome composition. In addition, there was no effect on biodiversity across environments and the major metabolic structure of the microbiome was stable across captivity status. However, we did observe changes in microbial community structure as sharks increased captive status, and only more specific (SEED subsystem: level 2 and 3) functional pathways varied within metagenomes.

### 4.1. Epidermal Microbiome Taxonomic Structure as a Product of Captivity Duration

Here, we show the composition of *T. semifasciata* populations differ in proportion of major contributors (>10% relative abundance) between captive and wild environments; we observed the Pseudoalteromonadaceae relative abundance to be significantly higher in captive samples, while their presence was only maintained in wild microbiomes. Consequently, we observed a shift in taxonomic compositions as *T. semifasciata* populations, including decreasing proportions of Pseudomonadaceae and Moraxellaceae, as Flavobacteriaceae and Pseudoalteromonadaceae became the dominant phyla in captive environments. The Pseudomonadaceae family, which belongs to the higher Gammaproteobacteria phylum and encompasses a diverse clade of microbes known to perform essential tasks such as nitrogen fixation and contaminant degradation. The observed change from microbiomes dominated by generalists (Pseudomonadaceae) in wild environments to opportunists (Pseudoalteromonadaceae, Flavobacteriaceae) in captive metagenomes may signify wild environments drive the selection of fast-growing associates by providing an abundance of resources [64,65]. Likewise, the increased relative abundance of the copiotrophic Pseudomonadaceae in wild groups may reflects unclean, near-shore conditions resulting from anthropogenic forces. The mostly aerobic Flavobacteriaceae family of microbes are physiologically similar to the Pseudomonadaceae and Pseudoalteromonadaceae families [66]. However, the increased relative abundance of Flavobacteriaceae in captive metagenomes may reflect a reduced available nutrient as microbes utilizing slow growth to navigate starvation naturally outcompete copiotroph rivals [67]. The Muricauda sp. likely benefit from the unsteady ocean environments they are cultured from, capitalizing on microbial turnover; the Muricauda sp. has been discovered in extreme conditions consisting entirely of a carbon source (hexadecane) it is not able to degrade, instead subsisting on cell lysis in the surrounding area. Much like the immune system of eukaryote that removes the buildup of cellular components as turnover occurs, this genera metabolizes the cellular debris in the unsteady state of the epidermal microbiome belonging to a host inhabiting an aquatic environment [68].

The epidermal microbiome alpha-diversity showed no significant differences across host environment, with less abundant taxa (<1%) accounting for little dissimilarity. Thus, we theorize the environment influences the relative abundance of the most abundant taxa while the hosts recruit taxon and maintain community diversity of epidermal microbiomes. Conversely, host factors such as biological sex, size, age, and pregnancy are known to play a role in shaping microbial associations the effect captive environments exert over major contributors to epidermal microbiomes suggests these and other host factors help maintain recurrent compositions that are worth exploring in future analyses. For example, the increased presence of Moraxellaceae in semi-captive sharks may reflect the transitional period from wild to captive lifestyles or a compounding effect of pregnancy as the semi-captive population was gravid during sampling. Therefore, while we have provided an important baseline representing the microbiomes of wild, semi-captive, and captive *T. semifasciata* populations, the gravid status of hosts may play a larger role in microbial community composition than the environment and individual host factors including age, sex, and weight should be investigated further.

### 4.2. Metabolic Potentials of Captive Shark Microbiomes Reflect Environmental Conditions

The epidermal microbiomes of captive *T. semifasciata* exhibited different metabolic potential although only at SEED subsystem level 2 and 3 and these higher-resolution variations are attributed to environmental differences. For example, captive *T. semifasciata* epidermal metagenomes had higher levels of several pathways involved in carbohydrate metabolism including mannitol utilization and fructose and mannose inducible phosphotransferase. While increased abundances of carbohydrate metabolism pathways in captive shark microbiomes are indicative of high levels of simple sugars in the environment, higher observed levels methylglyoxal pathway (MG) potential in captive metagenomes suggests excess nutrients available for captive microbes. The MG pathway produces aldehyde methylglyoxal instead of adenosine triphosphate, a toxic electrophile to cells. The MG pathway is therefore a low-energy-yielding glycolysis bypass that must be tightly regulated and is theorized to occur as a high-risk mechanism to harbor excess nutrients when catabolic rates increase, specifically sugar phosphate levels [69]. Correspondingly, alcohol synthesis pathways, i.e., acetone butanol ethanol synthesis gene pathway, metabolically follow carbohydrate metabolism and were also elevated in captive samples. Adjacent to nutrient metabolism, genes (SEED subsystems: Level 3) annotated for mycolic acid synthesis, peptide methionine sulfoxide reductase, and chorismite synthesis were all measured higher in captive metagenomes, indicating higher rates of cell wall synthesis, protein processing, and amino acid synthesis respectively. From these findings we surmise microbes associated with captive sharks benefit from stable abiotic conditions in aquaria, utilizing an overabundance of nutrients and experiencing increased catabolism compared to wild counterparts because of scheduled aquaria feeding and supplement offerings.

While captive metabolic pathways indicated the amount of environmentally available carbohydrates exceed wild environmental levels, limited heavy metal availability was equally inferred. Several specific, key functions found to be significantly higher in captive samples revealed captive environments to have iron limiting conditions, i.e., metabolic pathways such as iron acquisition, heme uptake and hemin utilization indicate a need for bacteria to sequester iron when the heavy metal is not readily found in the environment [70]. Competitive microbial interactions in captive environments could also be inferred from increased bacteriophage and vibrio-related genes such as murein hydrolases, i.e., hydrolytic enzymes known to cleave bacterial cell walls during final stages of the lytic cycle [71], and aspartate amino transferase, i.e., enzymes key for bacteriophage infection [72]. Last, shark microbiomes sampled in the captive environment displayed overall higher rates of protein misfolding within bacterial cell walls [73], as seen in periplasmic stress gene. Overall, captive environmental conditions prompt higher rates of nutrient-hoarding, phage interactions, and heavy-metal sequestering among microbes associated with *T. semifasciata* hosts. Further metagenomics studies should determine which factors besides captivity, e.g., temperature, light, etc, also promote these effects. In addition, while this data was collected from metagenomics, these findings could be further explored by metatranscriptomics.

### 4.3. MAGs reveal Novel, Constant Microbial Associations with T. semifasciata

Construction of genomes confirmed several microbial associations with *T. semifasciata* in addition to metabolic contributions these microbes encode. For one such association, we confirmed a novel and continuous relationship with two bacteria belonging to the *Muricauda* genera associated with *T. semifasciata* across captative environments. Historically, these genera of marine microbes are genomically underexplored and free living, isolated from both seawater and marine sediments in Antarctica [74,75,76,77] and more recently from the gills of shrimp in Okinawa [78]. Therefore, this is the first instance of the *Muricauda* genus being associated with a vertebrate host organism and may be a shark-specific microbe. The associative driving force between the *Muricauda* genus and Chondrichthyes is not immediately apparent: while other deep-sea, marine bacteria are capable of utilizing trimethylamine N-oxide (TMAO; a well-known organic compounds used by sharks and other fish for osmotic regulation), as a carbon and nitrogen source, the *Muricauda* genus was shown to lack the necessary enzymatic activity to reduce the macromolecule [79]. Although some members of the Bacteroidetes phylum are characterized as opportunistic pathogens, the *Muricauda antarctica* species is deemed a low pathogen risk.

Additional MAG annotation belonging to the *Bacteroidetes* phylum was Bin 21, identified as *Roseivirga pacifica*. The *Roseivirga* genera have been isolated from seawater and sea urchins (*Strongylocentrotus intermedius*) that inhabit the Pacific Ocean and are featured in benthic shark diets, thus possibly mediating the recruitment of *Roseivirga* genera to *T. semifasciata* [80]. The annotation of Bin 30 as *Zunongwangia atlantica* confirmed another microbe discovered in a deep-sea environment [81], and similar to *Muricauda* genera, are also non-motile and strictly aerobic. Bin 22, identified as *Leeuwenhoekiella blandensis*, similarly aerobic, prefers warmer growth conditions (28–30 °C; [82])and may be explained by aggregations of *T. semifasciata* in summer months.

While we were unable to annotate Bins 36 and 47 with confidence, the MAGs were found to contain the least contamination of the nine investigated MAGs. Uncharacterized MAGs are thought to be novel species as their sequences were highly complete. Future annotation of these genomes will no doubt reveal unique host-microbe associations and implications for host-microbe interactions. It should be noted the assembly of MAGs revealed a significantly larger contribution of contigs from wild samples, suggesting higher microbial loads in wild environments.

## 5. Conclusions

We investigated the associated microbial communities belonging to *T. semifasciata* housed since birth at the Birch Aquarium at Scripps Institution of Oceanography in La Jolla, a population held semi-captive (<1 year) at the Scripps institute of Oceanography, and wild populations in La Jolla and Moss Landing, California during summer months. We found these benthic shark epidermal microbiomes to be taxonomically distinct from respective environmental water columns with little interspecific variability. Our investigations into the impact of captivity on microbiome profiles revealed no significant differences in alpha-diversity indices and therefore captivity exerted no detrimental effect on biodiversity of shark-associated epidermal microbiomes.

We measured the similarity of epidermal microbiomes community compositions belonging to *T. semifasciata* across environments to be the same, albeit with varied structures, as beta-diversity measures showed no significant difference between the groups. We also discovered as sharks are held in captivity for longer durations, their microbiomes proportionally deviate from wild hosts; our results suggest captive environments influence relative abundances of key generalist bacteria while hosts regulate the presence of microbes. Last, metagenome-assembled genomes from *T. semifasciata* epidermal microbiomes identified and confirmed novel and consistent associations between the *Muricauda*, *Zunongwangia*, *Roseivirga* and *Leeuwenhoekiella* genera and the shark hosts. All groups contributed to the generation of the genomes (Figure 6), confirming the persistent presence of these microbes associated with *T. semifasciata*.

Baseline metrics for epidermal microbiomes belonging to *T. semifasciata* provide a foundation for wildlife research and conservation efforts. For example, local estuaries and bays have experienced massive die offs as a result of microbial blooms sequestering oxygen and pollution harboring disease-causing pathogens [7]. Early indicators of microbial dysbiosis via metagenomic analysis of associated microbiomes can provide aquariums with a non-invasive addition to their repertoire of environmental and animal monitoring. Already, mobile sequencing apparatuses are being deployed in the field to sequence without the use of a dedicated laboratory [83].

## Figures and Tables

**Figure 1 microorganisms-10-02081-f001:**
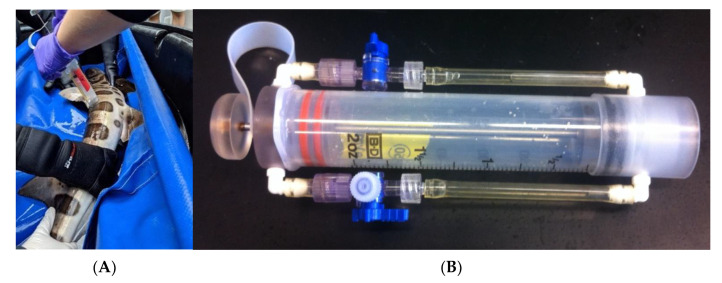
Sampling of the flank region on T. semifasciata (**A**) with a closed-circuit, blunt faced syringe tool (**B**) used for the collection of shark skin microbiome samples.

**Figure 2 microorganisms-10-02081-f002:**
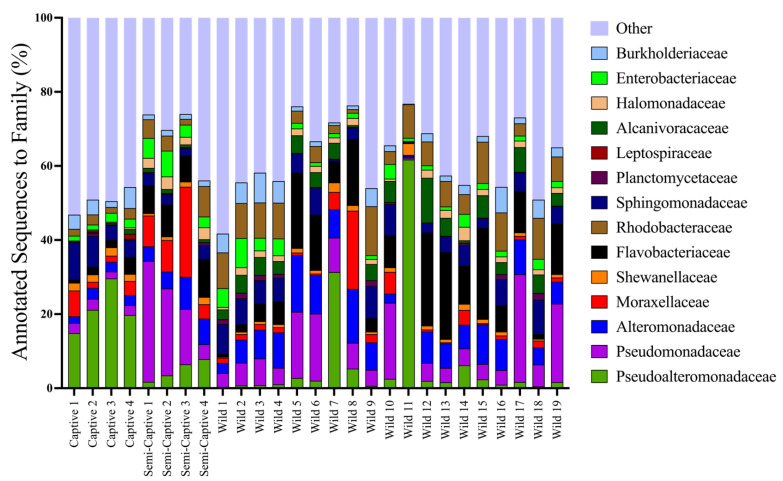
Variations among epidermal microbiome taxonomic composition belonging to *Triakis semifasciata* across captivity status. Height of bars represent relative abundance of most abundant phyla (>1%) at family level. The “Other” represents families that each have <11% proportional abundance.

**Figure 3 microorganisms-10-02081-f003:**
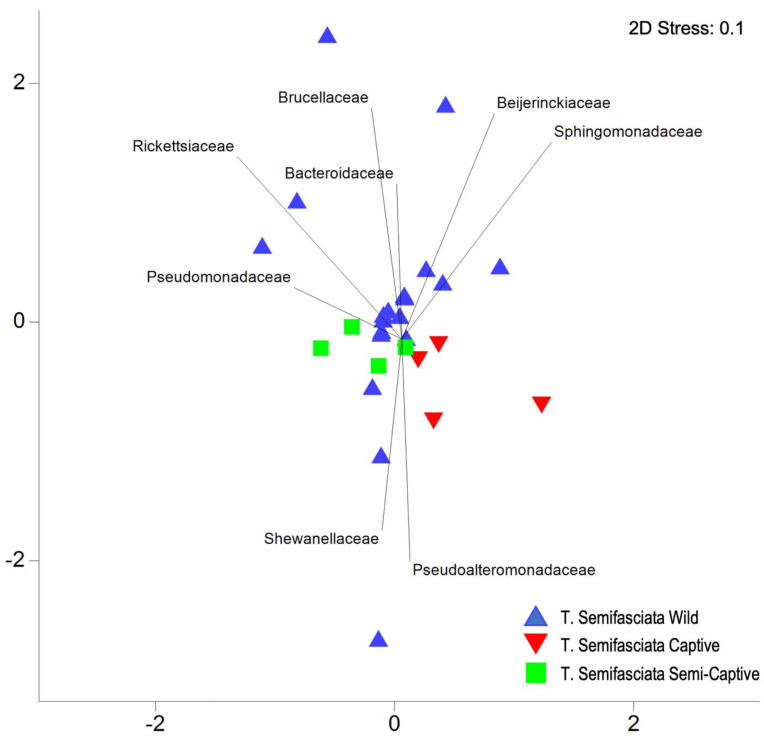
Non-metric dimensional scaling of Bray–Curtis similarity between epidermal microbiomes belonging to *Triakis semifasciata* across environments. Weak community dissimilarity observed between wild and captive populations at family level, with taxa influencing spatial groupings overlayed (>0.85 correlation). Relative abundance is normalized, fourth root transformed, and analysis is corrected with Bonferroni.

**Figure 4 microorganisms-10-02081-f004:**
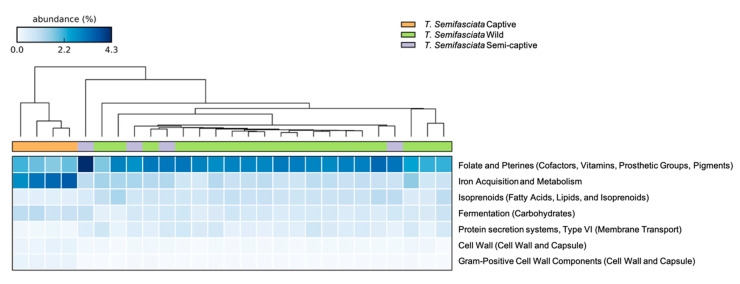
Significant differences in gene potentials (SEED Subsystem: Level 2) observed in epidermal microbiomes associated with T. semifasciata. Clustering dendrogram above heatmap orients nearest neighbor clustering of T. semifasciata metagenomes grouped by host environment and captivity status: long-term captive individuals (Birch Aquarium, blue), short-term captive individuals (Scripps Institute of Oceanography, purple), and wild individuals (green). Variations of relative functional gene abundances are visualized using a bi-colored heatmap from higher (blue) to lower (white) functional gene abundance.

**Figure 5 microorganisms-10-02081-f005:**
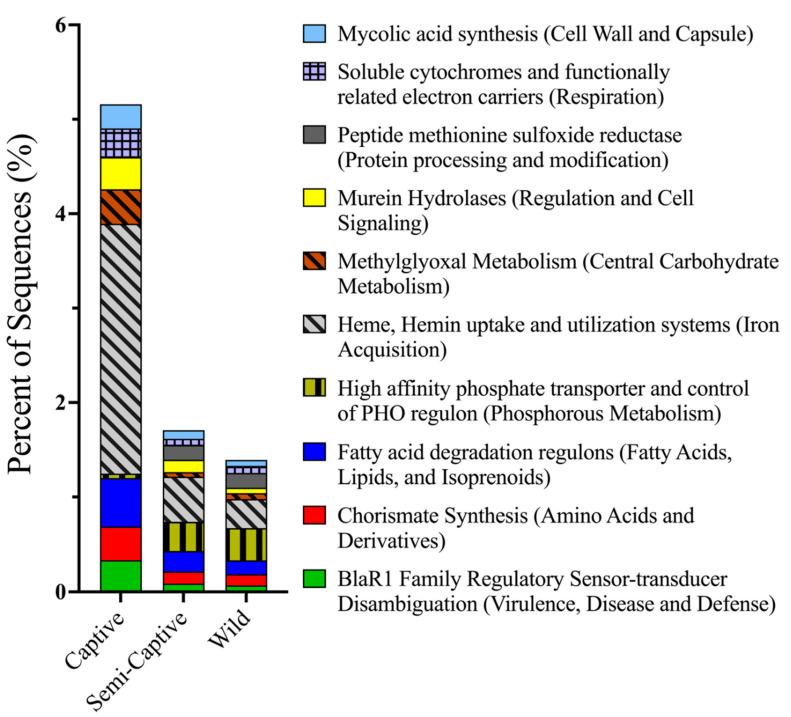
Percent distribution of functional pathways within epidermal microbiomes belonging to *Triakis semifasciata* across environments. Systematic top ten percent of reads representing specific (SEED subsystems: Level 3) gene pathways significantly differing between epidermal microbiomes belonging to captive, semi-captive, and wild *Triakis semifasciata* across environments.

**Figure 6 microorganisms-10-02081-f006:**
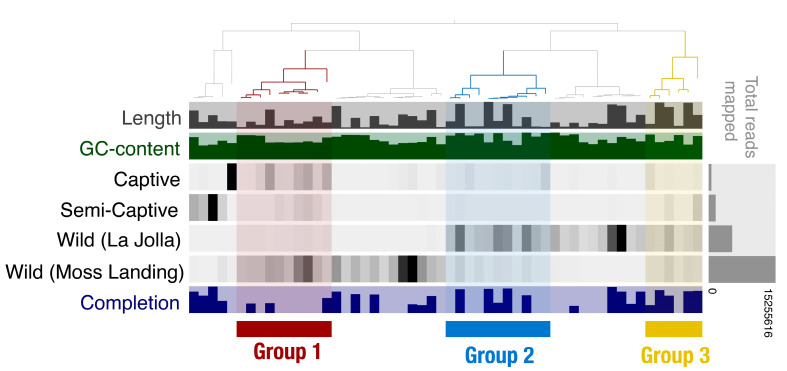
Phylogram depicting mean coverage of MAGs, ordered by total reads mapped. Highlighted groups aid visualization of even contribution across captive environments in three groups, and location specific MAGs in gray.

**Figure 7 microorganisms-10-02081-f007:**
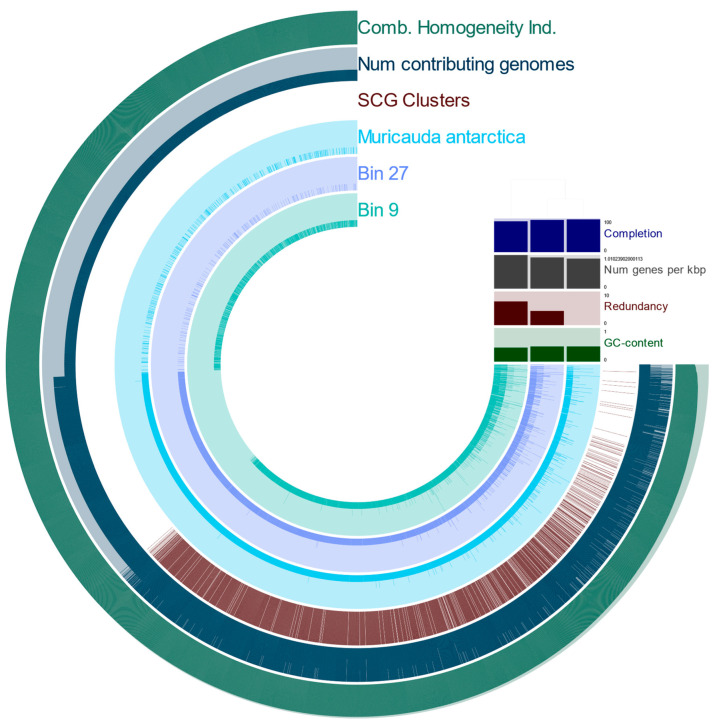
Pangenome frequency of Muricauda antarctica compared to MAG Bins 9 and 27. The clustering dendrogram to the right of the circular phylogram displays the hierarchical clustering of contigs based upon their differential sequence coverage across MAGs and the Muricauda antarctica reference genome (light blue). Below the dendrogram are genome indices including completion (%), number of contributing genes per 1 kbp for each open reading frame found, the redundancy level, and GC-content. Each sample layer (Muricauda antarctica, Bin 27, and Bin 9) represents gene cluster frequency, ordered by the combined homogeneity index auxiliary layer. Additional auxiliary layers from outside in, following combined homogeneity index, represent the number of contributing genomes and single copy gene clusters.

**Table 1 microorganisms-10-02081-t001:** Metadata and base pair and sequence counts for sampled *Triakis semifasciata* across environments.

Sample	Sex	Base Pair Count	Sequence Count
Captive 1	Male	13,254,680	36,040
Captive 2	Female	165,572,548	563,299
Captive 3	Female	129,410,990	454,887
Captive 4	Female	140,813,416	512,408
S.C. 1	Female	194,456,844	683,590
S.C. 2	Female	114,129,299	364,329
S.C. 3	Female	106,800,378	351,397
S.C. 4	Female	147,769,235	491,188
Wild 1	Female	263,295,773	1,298,868
Wild 2	Female	262,597,687	1,176,351
Wild 3	Female	258,939,703	1,226,960
Wild 4	Female	653,532,367	2,531,410
Wild 5	Female	640,031,260	2,434,754
Wild 6	Female	673,699,889	2,549,181
Wild 7	Female	397,391,784	1,368,629
Wild 8	Female	273,016,668	1,039,793
Wild 9	Female	291,042,619	1,045,530
Wild 10	Female	291,228,603	1,001,807
Wild 11	Female	369,123,866	1,254,908
Wild 12	Female	211,033,215	710,743
Wild 13	Female	279,527,632	914,034
Wild 14	Female	195,358,159	648,568
Wild 15	Female	109,131,187	328,208
Wild 16	Female	94,795,257	316,833
Wild 17	Female	186,632,228	623,111
Wild 18	Female	342,901,940	1,246,300
Wild 19	Female	217,511,438	709,664

**Table 2 microorganisms-10-02081-t002:** Average biodiversity indices for epidermal microbiomes belonging to *Triakis semifasciata* across environments.

Host Environment	Margalef’s (d) Index ± S.E.M. *	Pielou’s (*J’*) Index ± S.E.M.	Inverse Simpson (1/λ) Index ± S.E.M.
Captive	41.21 ± 4.15	0.59 ± 4.3 × 10^−2^	70.50 ± 3.54
Semi-captive	41.53 ± 3.54	0.624 ± 3.37 × 10^−2^	73.30 ± 2.27
Wild	40.07 ± 1.44	0.581 ± 2.88 × 10^−2^	68.29 ± 2.29

* S.E.M., Standard error mean.

**Table 3 microorganisms-10-02081-t003:** Pairwise SIMPER statistical comparison of the overall dissimilarity between the skin microbiomes belonging to *Triakis semifasciata* across environments and the corresponding microbes contributing ≥1% on average.

Scheme		
Family Level	% Dissimilarity	Contributing Microbes
All vs. Water	19.05	Moraxellaceae, Pseudomonadaceae, Rhodobacteraceae, Planctomycetaceae, Halomnocadaceae, Shewanellaceae, Enterobacteriaceae, Rickettsiales, Parachlamydiaceae, Cyanobacteria
Wild vs. Captive	17.4	Alteromondales, Pseudoalteromonadaceae, Rhodobacterales, Alcanivoraceae, Flavobacteriaceae, Caulobacteraceae, Erythrobacteraceae, Comamonodaceae, Alteromonoadaceae, Pseudomonadaceae, Rickettsiales, Halomonodaceae, Rhodobacteraceae
Wild vs. Semi-Captive	15.4	Moraxellaceae, Pseudomonodaceae, Rickettsiales, Pseudoalteromonadaceae, Rickettsiales, Alcanivoraceae, Erythrobacteraceae, Alteromonadales, Flavobacteriaceae, Xanthomonadaceae
Captive vs. Semi-Captive	13.4	Alteromonadales, Pseudomonadaceae, Pseudoalteromonadaceae, Halomonadaceae, Rhodobacterales, Flavobacteriaceae, Caulobacteraceae, Moraxellaceae, Bradyrhizobiaceae, Rickettsiales

**Table 4 microorganisms-10-02081-t004:** PERMANOVA and PERMDISP statistical comparisons of the effect of captivity on the composition and degree of in-group β dispersion of the skin microbiomes and the gene functions of *Triakis semifasciata*.

	PERMANOVA					PERMDISP	
Family Level	d.f.	Sum Sq	Mean Sq	Pseudo-*F*	*p*-(Perm)	*F*-Value	*p*-Value
Wild vs. Captive vs. Semi-Captive	2	446	223	1.84	0.054	1.29	0.598
Residual	25	2913.7	121.4				
Total	27	3359.7					
Genera Level							
Wild vs. Captive vs. Semi-Captive	2	1045.7	261.41	1.68	0.085	1.85	0.665
Residual	25	3886.2	155.45				
Total	27	4931.9					
Gene Function: Subsystem Level 2							
Wild vs. Captive vs. Semi-Captive	2	774.15	129	1.69	0.082	1.58	0.472
Residual	25	3048.2	76.21				
Total	27	3822.3					
Gene Function: Subsystem Level 3							
Wild vs. Captive vs. Semi-Captive	2	3712	618.7	1.79	0.052	4.02	0.068
Residual	25	13,848	346.2				
Total	27	17,560					

Note: d.f., degrees of freedom; Sum sq, sum of squares; Mean Sq, mean of squares; pseudo-*F*, *F*-value by permutation; *p*-values based on 999 permutations.

## Data Availability

All data and sequences were deposited in the NCBI Sequence Read Archive Database, under Accession Number PRJNA826531.

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
