# Peer review of "Epidermal Microbiomes of Leopard Sharks (Triakis semifasciata) Are Consistent across Captive and Wild Environments"

_microorganisms, 2022, doi:10.3390/microorganisms10102081_

Round 1

Reviewer 1 Report

This study looked at the microbial community of sharks in three environments, wild, semi-captive, and captive using shotgun metagenomics. The study compared these three environments to one another, however, the study design was unbalanced which likely led to misleading conclusions. The authors used 19 wild samples and compared these to 4 samples from the other two groups. In the conclusion, everything is more abundant in the captive animals and this makes sense because there were more samples in this dataset, but these results are unlikely biologically driven. The authors do not mention if they normalized genes at any point by the sequencing depth and thus larger libraries are also a bias in the statistical analysis. I would also like to see the authors use more metagenome-specific tools for differential abundance analysis such as MaAsLin2 to conduct normalization and transformations of count data. I think the authors need to redo all the analyses and use only 4 captive samples (from the closest timepoints to the wild and semi-captive samples) for statistical comparisons. Then the authors can use the rest of the captive samples to ask how captive samples change through time. 

Introduction

Line 44 I think the word reservoir should be changed to something like resident or microbial signatures. Reservoirs are more descriptive of pathogens and I don’t think that’s the only type of microbe the authors are referring to here.

Line 49 the word “thus” here is a bit out of place please rewrite this sentence

Line 58 citations of captive marine vertebrates are warranted here

Line 88 is there a reference for this or maybe reword or remove it as the opposite is unfortunately also true and should be stated.

Line 89 I also think “healthy living conditions” is too presumptive here. Something like potentially reducing pathogens 

Line 93 change from eukaryotes to sharks

Line 106-111 - The list of reasons why this is the model organism of choice doesn’t support the questions or design. I think only the last point “are successfully kept in display settings, unlike larger sharks” is relevant. I think the other points are interesting and relevant to mention just not as supportive of this being a good model for this study.

Line 123-129 Is this hypothesis based on any specific literature? It sounds like the authors did the analysis found the results and now are claiming this was their original hypothesis. There is not a clear indication from the reading why the authors think this. Please remove this or reword it. 

Lines 136-149 - This paragraph is difficult to read and difficult to get a clear picture of the sample size and dates. Please restructure or make a table of the sampling scheme. The data set is also unbalanced

Line 118-121 I don’t understand this sentence and think it may be unnecessary. 

Line 150-152 Fig 1 should be placed after this paragraph 

Line 154-156 I’m not sure the type of sample that’s being collected. Is it mucus, some epidermis cells? Was there scrapping with the blunt syringe or was the syringe moved up and down to create some pressure?

Line 157-159 I think this is an assumption and yet to be proven so should be written as such

Line 164 replace “again” with “then”

Line 172 remove “-“ in Mi-Seq

Line 196 the parenthesis was not closed

Line 211 - 215 Were these data transformed to take into account the compositionality of the data?

Line 215. How was alpha-diversity normalized and/or was the data checked for normality?

Lines 239-265 it is unclear how these data were normalized and how different sequencing depth was taken into account. 

Line 241 - 242 In the referenced study, Franzosa et al. 2014, the authors state that 41% of genes are not different from the transcript which means the majority (59%) are different and this to me does not support the claims that metagenomes can be used as a metatranscriptome proxy. In addition, the authors also show that beta-diversity were different between RNA and DNA.

Line 247 - was the data normalized by library sequencing depth how were the different sequencing depths addressed? 

Line 271 -missing period

Line 275 - what does “high” dissimilarity mean? What analysis and statistics were used.

Line 275 -278 This sentence is very unclear. How exactly is this different from the previous statement and what tests were used to arrive at this conclusion?

Line 289 - It’s debatable that family is a fine taxonomic level

Line 287-299 Should figure 2 be referenced here?

Line 308-310 A PCA plot should be paired with this result

Lines 283-299 and lines 310-324 are very similar in that they are both describing the taxonomic composition and should just be combined. 

For lines 308-310 and lines 324-327 were the same values used for PERMANOVA, PERMDISP and to generate the NMDS? The PERMANOVA, and PERMDISP don't align with Figure 3. The wild samples are clearly more dispersed than captive and semi-captive and there is clustering within captive and semi-captive. The variation within the wild samples may be explained by year and this should be depicted by shape. Nonetheless, the statics contradict the figure so the methods need to be revised. Also, the authors should conduct pairwise PERMANOVA, and PERMDISP comparisons. 

Line 471- 472 This is a bold statement especially given that there were differences reported throughout the text and in particular, lines, 474-476 contradict this statement.

Author Response

We would like to thank reviewer 1 for their assessment and valuable time. We have attached our response as a word document.

Reviewer 2 Report

The paper entitled Epidermal Microbiomes of Leopard Sharks (Triakis semifasciata) Are Consistent Across Captive and Wild Environments have a main purpose to investigate the impact that a captive environment exerts on the epidermal microbiota associated with captive T. semifasciata individuals and compare these microbial communities to the composition and functional potentials of microbiomes belonging to wild sharks that live in nearshore habitats and temporarily held sharks representing semi-captivity. The authors also hypothesize the taxonomic composition of the epidermal microbiomes belonging to T. semifasciata will remain consistent between captive and semi-captive sharks and wild peers. Furthermore, while the broad functional genes are expected to remain largely unchanged across captivity status, a few key functions are anticipated to differ as a reflection of the provisioning that occurs in captivity. Last, by showed results the authors identify the presence of novel microbes that are consistent across wild, semi-captive, and captive sharks by constructing metagenome assemble genomes.

The paper is structured in five parts excepting abstract and references. In part one, Introduction are presented the state of the art into investigated field.

The part two is structured in four chapters in which are detailed the methods used for the obtaining the results according to the goals of the study. In part tree, divided in two subchapters is described in detail the obtained results. This part is detailed in two subchapter in which are described and comment the results. Chapter four is dedicated to the discussion of the results into frame of the state of the art in the investigated field and contain tree subchapters.

As major conclusion, the authors argued baseline metrics for epidermal microbiomes belonging to T. semifasciata provide a foundation for wildlife research and conservation efforts. For example, local estuaries and bays have experienced massive die offs as a result of microbial blooms sequestering oxygen and pollution harboring disease-causing pathogens. Early indicators of microbial dysbiosis via metagenomic analysis of associated microbiomes can provide aquariums with a non-invasive addition to their repertoire of environmental and animal monitoring. Already, mobile sequencing apparatuses are being deployed in the field to sequence without the use of a dedicated laboratory.

Hharks are held in captivity for longer durations, their microbiomes proportionally deviate from wild hosts; our results suggest captive environments influence relative abundances of key generalist bacteria while hosts regulate the presence of microbes. Metagenome-assembled genomes from T. semifasciata epidermal microbiomes identified and confirmed novel and consistent associations between the Muricauda, Zunongwangia, Roseivirga and Leeuwenhoekiella genera and the shark hosts. All groups contributed to the generation of the genomes confirming the persistent presence of these 614 microbes associated with T. semifasciata.

Considering the quality of the manuscript I recommended to be accepted for publication without further modifications.

Author Response

Point 1: Considering the quality of the manuscript I recommended to be accepted for publication without further modifications.

Point 1: We would like to thank reviewer 2 for their assessment and valuable time. 

Reviewer 3 Report

Congratulations. Very impressive research !!!! 

The topic is very interesting and very very original, quite suitable for the journal. Indeed,  the mkcrobiome of the skin plays a vital role in the health and the survival of marine organisms. It not only reflects the environmental pressures but also protects the host. The authors describe the importance of the epidermal microbiomes in the introduction with clarity. The question under research is wheteher the captive environment affects the epidermal microbiome of the leopard sharks in comparison to that of the wild sharks. 

The experimental design is suitable to adress the research question (sampling and metagenomes analysis).

The results are presented with clarity. Figures, graphs and tables help to understand the text, which is very technical.

The conclusions are based on the results and follow their logic. Captive environments are important for conservation and education purposes if and only if are safe for the captured animals. Captivity duration affected the epidermal microbiome and the metabolic potentials of the captive dhark microbiomes reflected environmental conditions .For these reasons the monitoring of epidermal metagenomes of captive species may help the early doagnosis of diseases and save lives.

A very very useful research.

Author Response

We would like to thank reviewer 3 for their assessment and valuable time.

Reviewer 4 Report

I really appreciated this paper, which deals with a topic of extreme topicality and scientific interest. The study of the skin microbiome of elasmobranchs, in fact, is a recent line of research that needs contributions like this.

The paper is really well structured and very clear to understand, even by readers who do not have many skills in this regard.

Only at line 153 it would be appropriate to specify how the water was filtered to be decontaminated before being used for the extraction of the microbiome from the skin of the sharks.

Author Response

Point 1: Only at line 153 it would be appropriate to specify how the water was filtered to be decontaminated before being used for the extraction of the microbiome from the skin of the sharks.   

Point 1: To clarify for reviewers and readers the filtration process for seawater, the following has been added to line 153: “The syringe is prefilled with the seawater passed through a filter with a nominal molecular weight cut-off of 100kDa that was then flushed over the epidermis to dislodge microbes without introducing microbes from the environment”.

We would like to thank reviewer 4 for their assessment and valuable time.

Round 2

Reviewer 1 Report

Major Point 1: The authors do not mention if they normalized genes at any point by the sequencing depth and thus larger libraries are also a bias in the statistical analysis. The following has been added to improve upon point 1 to line 224: 

“To produce relative abundance of each taxon level, the data sets were normalized to the sum of all taxa counts for each epidermal microbiome sample… All data was fourth root transformed [4] which balances the effects of a community structured on a few abundant species and a community structured on all species, and thereby influenced by the occurrence of the rarest taxa [51].” 

Re Major Point 1: This method does not address sequencing depth or the imbalance in the dataset. The captive samples have more abundant genes because there are more captive samples. 

Major Point 2: I would also like to see the authors use more metagenome-specific tools for differential abundance analysis such as MaAsLin2 to conduct normalization and transformations of count data. We look forward to exploring this tool for future papers.

 For the purpose of this work, normalization and transformation was accomplished using in-software tools found in excel, PRIMER, and Graphpad which are routinely used to handle large data sets. 

Re Major Point 1: The programs mentioned do not consider the difference in gene length and the data should be normalized using something like RPKM and then tested with metagenome-specific tools which account for the compositionality of the data and should be applied to this study.

Major Point 3: I think the authors need to redo all the analyses and use only 4 captive samples (from the closest timepoints to the wild and semi-captive samples) for statistical comparisons. Then the authors can use the rest of the captive samples to ask how captive samples change through time. 

To satisfy this suggestion we have randomly selected four wild samples and compared these to the captive groups ten times. Below is a table of the results followed by a discussion.

This analysis is confusing to me. What was the input data and how was it transformed? It would be helpful if the authors would make their code publically available or include it as supplemental material. Regardless, the year and higher number of captive samples is a confounding factor that should be accounted for in the statistics and visually explored. 

  • Line 316 Where does citation 30-32 mention rarefaction? In addition, relative abundance is not a method to normalize for richness, and evenness. The input values should be discrete count data and not percentages. Richness is not affected by relative abundance as you are counting the number of taxa so this does not normalize the data. For evenness, the data is transformed into relative abundances within the formula, so it is not necessary to transform the data, but it is necessary to normalize by sequencing depth.
  • Line 387 This does not measure which genes are important in an environment it tells you which genes are present in an environment
  • Line 387 There is “some” level of correlation not high. Cite the actual numbers.
  • Line 392-394 Please remove this sentence. While it is true that mRNA is more stable mRNA provides a better proxy of function and extracting mRNA from marine samples is feasible and does not need to be replaced by DNA.
  • Line 399 - 411 As mentioned in the first review this data should be normalized and relative abundance is not a form of normalizing sequence alignments. The coverage of each metagenome should be normalized which can be done by CoverM which normalizes metagenomes by library size using the trimmed mean function. However, this still doesn’t account for the imbalanced dataset.
